# Risk of Metabolic Syndrome in Kidney Stone Formers: A Comparative Cohort Study with a Median Follow-Up of 19 Years

**DOI:** 10.3390/jcm10050978

**Published:** 2021-03-02

**Authors:** Robert M. Geraghty, Paul Cook, Paul Roderick, Bhaskar Somani

**Affiliations:** 1Department of Urology, Freeman Hospital, Newcastle-upon-Tyne NE7 7DN, UK; rob.geraghty@newcastle.ac.uk; 2Department of Biochemistry, University Hospital Southampton, Southampton SO16 6YD, UK; Paul.Cook@uhs.nhs.uk; 3Department of Public Health, University of Southampton, Southampton SO16 6YD, UK; pjr@soton.ac.uk; 4Department of Urology, University Hospital Southampton, Southampton SO16 6YD, UK

**Keywords:** kidney stones, metabolic syndrome, urolithiasis, nephrolithiasis, kidney calculi, diabetes mellitus

## Abstract

Background: Kidney stone formers (SF) are more likely to develop diabetes mellitus (DM), but there is no study examining risk of metabolic syndrome (MetS) in this population. We aimed to describe the risk of MetS in SF compared to non-SF. Methods and Materials: SF referred to a tertiary referral metabolic centre in Southern England from 1990 to 2007, comparator patients were age, sex, and period (first stone) matched with 3:1 ratio from the same primary care database. SF with no documentation or previous MetS were excluded. Ethical approval was obtained and MetS was defined using the modified Association of American Clinical Endocrinologists (AACE) criteria. Analysis with cox proportional hazard regression. Results: In total, 828 SF were included after 1000 records were screened for inclusion, with 2484 age and sex matched non-SF comparators. Median follow-up was 19 years (interquartile range—IQR: 15–22) for both stone formers and stone-free comparators. SF were at significantly increased risk of developing MetS (hazard ratio—HR: 1.77; 95% confidence interval—CI: 1.55–2.03, *p* < 0.001). This effect was robust to adjustment for pre-existing components (HR: 1.91; 95% CI: 1.66–2.19, *p* < 0.001). Conclusions: Kidney stone formers are at increased risk of developing metabolic syndrome. Given the pathophysiological mechanism, the stone is likely a ‘symptom’ of an underlying metabolic abnormality, whether covert or overt. This has implications the risk of further stone events and cardiovascular disease.

## 1. Introduction

Kidney stone disease (KSD) is a costly [1] and increasingly prevalent problem, with the latest USA prevalence (2015–2016) being 10% [2]. Amongst the risk factors for development of KSD, type 2 diabetes mellitus and the metabolic syndrome (MetS) [3] are particularly well described. Both are characterised by high blood glucose and insulin resistance [4] and share common pathophysiologic mechanisms that attributes to the increased risk of KSD, e.g., urinary acidification [5]. This translates to a proportional increase in uric acid stones [6]. Given the MetS pandemic [7], this will lead to worldwide increases in KSD.

The other components of MetS (obesity, hypertension and dyslipidaemia) have all been described, to varying degrees, as carrying increased risk of KSD. There is good epidemiological evidence for the link between obesity and an associated risk of KSD [8]. However, the cause of this increased risk is likely due to the metabolic sequelae of obesity, such as dyslipidaemia and insulin resistance [5].

There is conflicting evidence for the risk of KSD in hypertensives. Unadjusted crude risk demonstrates significantly increased risk of hypertensives becoming stone formers [9,10]. However, on adjustment the increased risk is rendered non-significant [9,10]. This is likely due to the confounding presence of other KSD risk factors, e.g., high blood glucose or dyslipidaemia.

Dyslipidaemia (high serum triglycerides and low high-density lipoprotein) causes demonstrable derangements in 24 h urinary biochemistries [11]. A further sequela of dyslipidaemia is lipotoxicity (abnormal lipid accumulation in tissues) [12]. In the kidney, lipotoxicity reduces ammonium secretion and lowers pH (both risk factors for KSD) [11].

Not only are the components of MetS risk factors for KSD, the reverse is also true. Stone formers are at increased risk of developing both diabetes mellitus [13] and hypertension [14]. As yet however, there is no evidence for increased risk of MetS in stone formers.

The importance of a MetS diagnosis is the increased risk of cardiovascular disease [15]. Although the definition has changed over the years, the consensus across multiple large cohort studies is a 2- to 4-fold increase in risk of cardiovascular disease in those with MetS. This has clinical implications for individuals and populations.

As there has been no study examining the risk of developing MetS in the stone forming population, our primary aim was to describe this risk in stone formers. Our secondary aims were to examine the risk of individual MetS components and risk of MetS per stone type.

## 2. Methods

### 2.1. Definitions

Metabolic syndrome was defined using the Association of American Clinical Endocrinologists (AACE) criteria [4], which is similar to the more widely used National Cholesterol Education Program Adult Treatment Panel (NCEP ATP) III criteria. It replaces waist circumference with (body mass index) BMI > 25, as waist circumference was not available on electronic records. In addition to AACE criteria, specific treatment for hyperglycaemia or hypertension were included, as well as physician diagnosis (see Table 1). Development of three or more components was defined as incident Metabolic syndrome (MetS). Age of development of MetS defined as age at which 3 or more components present, components assumed to be cumulative, i.e., patients will not lose diabetic or hypertensive, etc., status with increasing age.

Electronic records included all clinical letters, operation notes, test results, diagnoses, treatments, and basic readings including blood pressure, height, and weight.

### 2.2. Study Population

The cohort consisted of patients with kidney stone disease (KSD) presenting to a tertiary referral hospital referred for metabolic assessment between 1990 and 2007. The study population has been described in a previous cross-sectional study [16] and subsequent cohort study [1]. During this period, stone formers were routinely referred to this clinic by the urology team (both in Southampton and around the region—Dorset, Wiltshire, and Hampshire) and general practitioners. In total, 1000 (from 2801) patients were selected by block randomization after alphabetization of surnames.

Further information on past medical history and subsequent stone recurrence was ascertained retrospectively using hospital and general practice electronic records. The general practice electronic records is downloaded to the Care and Health Information Exchange (CHIE), a large database including data from 172 general practices within Hampshire and the Isle of Wight (95% coverage).

Data collected in retrospect using CHIE: age, sex, past medical history at first presentation, including metabolic syndrome components (see Table 1) and incident metabolic syndrome components. Subsequent stone episodes and stone type were ascertained using a combination of CHIE and hospital records. See Appendix A for stone disease read codes.

Patients who had no documentation (i.e., no evidence of subsequent follow-up or consultation, lived outside or have left Hampshire, or no documentation on CHIE) or had pre-existing metabolic syndrome (MetS) were excluded (see Figure 1).

### 2.3. Comparator Population

Comparator data was supplied by Care and Health Information Analytics (CHIA), the body utilising CHIE data for research, using age (within 5 years), sex, and region matched patients in a ratio of 3:1 once stone formers (SF) had been screened for eligibility. The follow-up period was matched as closely as possible.

Patients with codes associated with KSD (see Appendix A) and previous components of metabolic syndrome were excluded. Data on incident metabolic syndrome components were collected (see Table 1), time defined as initial age to age at which first reached diagnostic criteria for metabolic syndrome component.

Only practices which were present within CHIA on 1st May 2019 were selected to be included. Random patients were selected from this practice cohort. Data on age of development of MetS components and death (if applicable) were extracted.

### 2.4. Statistical Methods

SPSS (version 26, IBM, Armonk, NY, USA) and R statistical package version 3.6.3 (R Foundation for Statistical Computing, Vienna, Austria. URL https://www.R-project.org/) (packages: survival and survminer) were used for statistical analysis. Cox proportional hazards model was used to analyse the data, which is presented as hazard ratio (HR) with 95% confidence interval (CI). Time to event was defined as time from presentation to metabolic stone clinic to development of 3 components of metabolic syndrome for both stone formers and comparators. Censoring time was defined as time from presentation to metabolic stone clinic to last CHIE entry or death. We tested the proportional hazards assumptions by calculating Schoenfield residuals and performing a log-rank test.

Subanalyses for 0 and 1 or 2 previous components, as well as stone type. The main outcome measure was adjusted for number of previous components. Chi-squared test was used to compare prior to year 2000 vs. 2000 onwards for components of MetS.

### 2.5. Sample Size Calculation

Sample size was calculated estimating a 10% difference (30:40%) in rates of MetS diagnosis between the two groups. Power was set at 80% and significance at 0.05. Sample size was therefore calculated at *n* = 172 per group. Larger numbers have been included to increase power for subanalyses. The 3:1 ratio of controls to cases was used to increase robustness and power.

### 2.6. Ethical Approval

Ethical approval for this study was granted by the NHS Bristol Research Ethics Committee (Research ethics committee reference: 18/SW/0185; IRAS ID: 240061).

## 3. Results

### 3.1. Demographics

There were 828 stone formers and 2484 stone free comparators, with no differences in age or sex between the groups. Stone formers underwent a median 19 years (IQR: 15–22) of follow-up from initial presentation to biochemical clinic. Non-stone formers had data available for the same time period (median 19 years, IQR: 15–22).

There were 361 (43.6%) stone formers who developed metabolic syndrome (MetS), whilst 617 (24.8%) of the stone free comparators developed MetS. Numbers of components and primary stone composition are detailed in Table 2. Deaths were similarly proportioned in the two groups with 113 (13.6%) amongst stone formers, and 366 (14.7%) amongst the comparators.

There were 719 (86.8%) and 2118 (85.3%) stone free comparators without any prior components of MetS. There were 111 (13.4%) and 332 (13.4%) stone free comparators with 1 or 2 components.

### 3.2. Risk of Metabolic Syndrome in Stone Formers

Stone formers were at significantly increased risk of developing MetS (HR: 1.77; 95% CI: 1.55–2.03, *p* < 0.001) (see Figure 2 and Table 2). This effect was robust to adjustment for presence of previous components (HR: 1.91; 95% CI: 1.66–2.19, *p* < 0.001). This effect was consistent with subanalyses of no previous components (HR: 1.98; 95% CI: 1.69–2.31, *p* < 0.001) and 1 or 2 previous components (HR: 1.54; 95% CI: 1.11–2.14, *p* = 0.011).

Subanalysis of stone type demonstrated significantly higher risk for stone patients compared to their matched comparators presenting with calcium oxalate (HR: 1.82; 95% CI: 1.53–2.16, *p* < 0.001) and urate stones (HR: 3.87; 95% CI: 2.23–6.72, *p* < 0.001) (see Table 1). Other stone types did not carry significant risk of developing MetS.

Subanalysis of individual components of the metabolic syndrome demonstrated SFs were significantly more likely to develop all bar impaired glucose tolerance on both unadjusted and adjusted analyses (see Table 3). Those with the component pre-existing were excluded.

Numbers of patients at follow-up times were as follows: 5-years (control, *n* = 2484; SF, *n* = 828), 10-years (control, *n* = 2481; SF, *n* = 827), 15-years (control, *n* = 1938; SF, *n* = 646), 20-years (control, *n* = 1119; SF, *n* = 373), 25-years (control, *n* = 366; SF, *n* = 122).

There were significantly more patients with previous components of the metabolic syndrome after 2000 than prior (Chi-square, *p* < 0.001), despite this analysis of only those presenting after 2000 still had a significantly increased risk of developing MetS (HR: 2.42, 95% CI: 2.01–2.92, *p* < 0.001). Log rank demonstrated a significant result (*p* < 0.001). Visual inspection of the Schoenfeld residuals did not demonstrate variation around 0, although it did demonstrate a significant result (global Schoenfeld test, *p* < 0.001) (see Figure 3). 

## 4. Discussion

This is the first study to examine the risk of metabolic syndrome in stone formers. There was a significant risk (nearly twice as likely) of developing metabolic syndrome in this population, which was more common still in those with uric acid stones.

The main strength of this study is an appropriately powered, significant primary outcome, which is robust to adjustment for previous components. The use of 3:1 matching of study participants to comparators for age and sex, improves power and robustness. Broadly, the sensitivity analyses (log-rank, Schoenfeld residuals and subanalyses) demonstrate results in keeping with the primary outcome.

The major limitation of this study is the risk of under-ascertainment of MetS at baseline (there were only 20 stone formers with MetS), this is reflected in significantly lower MetS components prior to 2000 in both groups. Routine screening of metabolic syndrome components by General Practitioners was not established until after the millennium, which would account for the previously mentioned observation. One would expect a higher number of stone formers to have pre-existing MetS, given that they are more likely to develop KSD [3]. However, the risk of under-ascertainment is likely to be inherent to both groups. We have also adjusted for prior components for both groups, and performed subanalyses on development of MetS with 0, 1, and 2 previous components. All of these analyses demonstrate highly significant results, increasing the likelihood that stone formers are indeed at increased risk of MetS.

There are several other weaknesses to this study. Firstly, the dataset used, Care and Health Information Exchange (CHIE) uses data inputted by general practitioners. Primary care data are known to be more variable and less accurate than secondary care data [17]. It was also not possible to match patient’s address’ and GP practice’s and therefore we were unable to adjust for deprivation. However, the expected results are significant (i.e., urate stones increase risk of MetS and increased risk of recurrence in stone formers with MetS), and therefore there is no risk of type 2 error. Secondly, risk of type 1 error may be present given the multiple testing in the secondary outcomes, and larger studies are needed to corroborate these findings. Lastly, there may be an argument that stone referrals to a tertiary referral service are not representative of the general stone forming population. However, the recurrence rate is similar to previously documented series (around 40% at 10 years in this cohort) [18], only a small proportion were started on prophylactic medication (16%) and there were similar ratios of stone types to previous series [6,19]. Due to these reasons, we believe this dataset is representative.

The increased risk of MetS in the stone forming population is significant given the rising prevalence of kidney stone disease (KSD), which was 10% in 2015–2016 in USA [2]. This translates into 38.2 million Americans who have had a kidney stone and are therefore at roughly twice the risk of developing MetS, with the associated 2- to 4-fold increased risk in cardiovascular disease [15]. It is should be noted it is unlikely to be all stone formers who develop MetS as there are alternative causes of KSD (genetic, infection, drugs, etc.) that have no association with MetS or cardiovascular disease [20].

It is clear that insulin resistance and renal lipotoxicity are the main drivers of stone formation in the MetS population [11,21]. However, it is not clear why stone formers are at increased risk of developing MetS. Our observation that stone formers are more likely to develop MetS correlates with previous studies on the increased risk of developing diabetes [13] and hypertension [14] in stone formers. Both MetS and DM are characterized by insulin resistance, which leads to urinary acidification and increased uric acid excretion [6,22] with a resulting higher proportion of uric acid stones [23]. Hypertension is also associated with urinary acidification along with hypocitraturia [24], both risk factors for stone formation. However, there is no evidence that kidney stones, or abnormalities in 24 h urinary biochemistry influence the development of MetS or its components.

Intriguingly, the link between KSD and MetS is reflected in the genetics literature. In genome wide association studies, two single nucleotide polymorphisms (rs780093 and rs1260326) within a single gene (*GCKR*) are associated with both KSD [25,26] and MetS [27]. This gene encodes glucokinase regulator protein, which is mainly expressed in the liver [28]. Although not yet demonstrated in functional studies, clinically these variants are associated with higher triglycerides and higher fasting plasma glucose [29], both of which are components of MetS and risk factors for KSD. KSD is therefore likely to be a result of metabolic derangements, given the association with these variants (no renal expression of *GCKR)* and the associated risk of KSD with higher triglycerides, higher fasting plasma glucose and MetS.

If KSD is indeed a symptom of an underlying metabolic derangement, rather than vice versa, then there may be evidence of metabolic dysfunction at presentation. It is unclear in the literature whether there is evidence of insulin resistance or renal lipotoxicity, or its surrogates (dyslipidaemia or high BMI) at this point, and we have discussed the risk of under-ascertainment of MetS components earlier. Interestingly, Sagesaka et al. demonstrated that type 2 diabetes could be predicted up to 10 years before the patient developed the condition using the same factors used to diagnose metabolic syndrome [30]. Unfortunately, they did not examine if the components of MetS rose and fell, respectively, as fasting plasma glucose did.

Futures studies should examine the presence of metabolic syndrome components in stone formers prospectively, examining risk of recurrence with metabolic syndrome and development of metabolic syndrome. The involvement of geno- and phenotype correlations should be considered. Preventative measures for both recurrent stones and components of metabolic syndrome should be trialled. More work also needs to be done on primary prevention and effect on patients quality of life [31,32].

Routine assessment for components of MetS should be standard when assessing a stone formers given the further risk of KSD and, perhaps more importantly, the long-term cardiovascular implications [15].

## 5. Conclusions

Kidney stone formers are at increased risk of developing metabolic syndrome, which is commoner with uric acid stones. A stone is likely a ‘symptom’ of an underlying, perhaps covert, metabolic derangement in idiopathic stone formers given the described pathophysiology.

This increased risk has both individual and health policy implications given the associated cardiovascular outcomes. Assessment for metabolic syndrome should be standard for patients presenting with kidney stones.

## Figures and Tables

**Figure 1 jcm-10-00978-f001:**
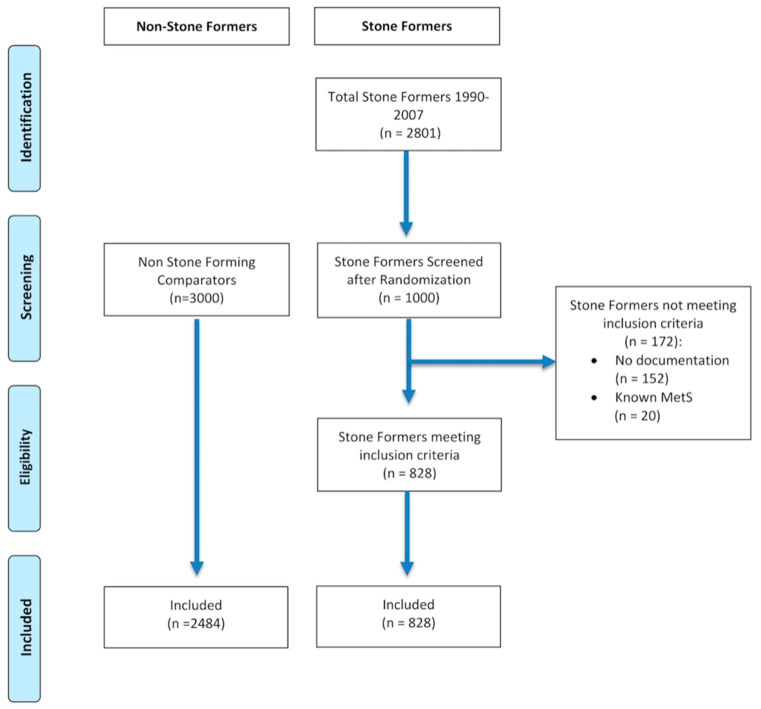
CONSORT flow diagram of patient selection.

**Figure 2 jcm-10-00978-f002:**
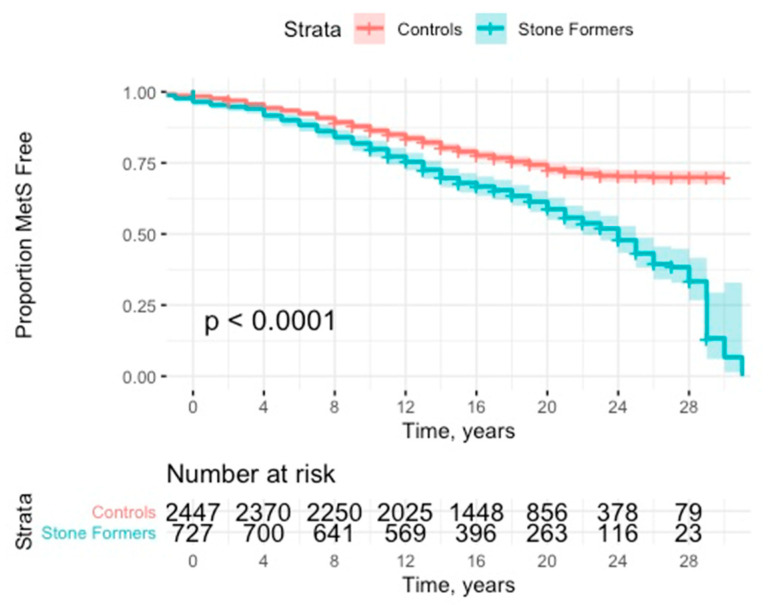
Kaplan Meier curve with 95% CI (confidence interval) for time to development of metabolic syndrome.

**Figure 3 jcm-10-00978-f003:**
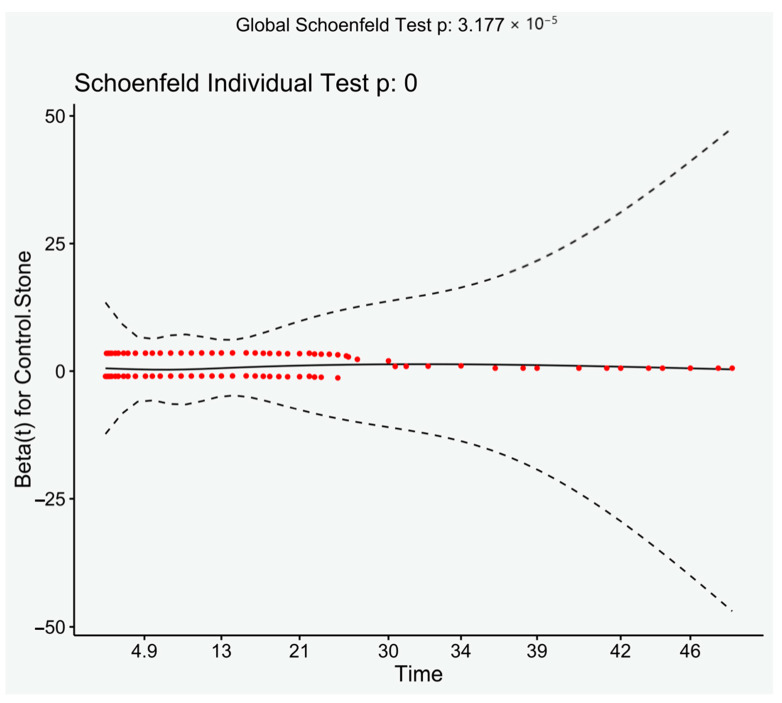
Schoenfeld residuals plotted against time. Loess line with 95% CI.

**Table 1 jcm-10-00978-t001:** Metabolic syndrome definition.

Metabolic Syndrome (Modified AACE Criteria)	
Fasting Plasma Glucose	>6.1 mmol/L or Hypoglycaemic treatment or Physician diagnosis of Impaired Glucose Tolerance or Diabetes Mellitus
Body Mass Index	≥25 kg/m^2^ or Physician diagnosis of Obesity
Blood Pressure	≥130/≥85 mmHg or Antihypertensive treatment or Physician diagnosis of hypertension
Triglycerides	>1.7 mmol/L
High-Density Lipoprotein	M: <1.04 mmol/L; F: <1.29 mmol/L

**Table 2 jcm-10-00978-t002:** Demographics of stone formers and stone-free comparators.

		Controls	Stone Formers	HR (95% CI)	*p*
Age at Presentation (Years), Mean ± SD		49 ± 14	49 ± 14		
Sex, *n* (%)	Female	723 (29.1%)	241 (29.1%)		
	Male	1761 (70.9%)	587 (70.9%)		
Follow-Up (Years), Median (IQR)		22 (17–27)	22 (17–27)		
Metabolic Syndrome, *n* (%)		617 (24.8%)	361 (43.6%)	1.77 (1.55–2.03)	<0.001
Metabolic Syndrome Components Developed, *n* (%)	0	478 (19.2%)	114 (13.8%)		
	1	793 (31.9%)	146 (17.6%)		
	2	596 (24.0%)	172 (20.8%)		
	3	399 (16.1%)	170 (20.5%)		
	4	182 (7.3%)	134 (16.2%)		
	5	36 (1.4%)	83 (1.0%)		
Primary Stone Composition, *n* (%)	Ca Ox	-	425 (51.3%)	1.82 (1.53–2.16)	<0.001
	Urate	-	21 (2.5%)	3.87 (2.23–6.72)	<0.001
	Ca Po	-	17 (2.1%)	0.89 (0.33–2.38)	0.82
	Struvite	-	5 (0.6%)	0.78 (0.11–5.54)	0.80
	Unclear	-	360 (43.5%)	1.71 (1.43–2.05)	<0.001

**Table 3 jcm-10-00978-t003:** Individual components of metabolic syndrome and overall risk. Adjusted for age and sex.

Component	Unadjusted		Adjusted	
	HR (95% CI)	*p*	HR (95% CI)	*p*
Impaired Glucose Tolerance	1.19 (0.97–1.46)	0.09	1.17 (0.95–1.43)	0.13
Hypertension	1.56 (1.41–1.81)	<0.001	1.51 (1.33–1.71)	<0.001
BMI > 25	1.41 (1.03–1.26)	0.01	1.11 (1.01–1.24)	0.04
TGL > 1.70	1.58 (1.37–1.83)	<0.001	1.50 (1.30–1.74)	<0.001
HDL < 1.04 for women; <1.29 for men	1.26 (1.09–1.45)	<0.001	1.25 (1.09–1.44)	0.002
Metabolic Syndrome	1.78 (1.56–2.03)	<0.001	1.77 (1.55–2.03)	<0.001

## Data Availability

As data is identifiable it will not be made available as per ethical approval.

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
