# Peer review of "Risk of Metabolic Syndrome in Kidney Stone Formers: A Comparative Cohort Study with a Median Follow-Up of 19 Years"

_jcm, 2021, doi:10.3390/jcm10050978_

Round 1
Reviewer 1 Report
In this study, the authors presented the risk of metabolic syndrome in kidney stone formers. Of the patients who visited their stone center, 828 were selected for the study. The control group consisted of 2484 individuals matched for age, sex, region, and follow-up period from CHIE (Care and Health Information Exchange) data. Authors reported that 43.6% of Stone formers developed metabolic syndrome, with a hazard ratio of 1.77 compared with controls. Patients with uric acid stones had a particularly high risk of developing metabolic syndrome.
The findings in this manuscript are important for the management of stone formers.
The manuscript is well written and well structured. However, I have some comments to report:
- As the authors noted, the most critical issue is that it is inaccurate to collect a component of MetS at the baseline. In their study, only 8.6% of the stone former had components of MetS (obesity, hypertension, dyslipidemia, or Impaired Glucose Tolerance). However, according to recent literature, more than 20% of patients with renal stones are reported to have hypertension (Ref 13 and 14 in their study). Obesity is common in patients with nephrolithiasis. They claimed that there was a risk of under occlusion in the control group. However, it cannot deny the possibility that many patients already had the component of MetS in Stone former at the start of the observation. If screening for MetS in GPs has been expanded after 2000, it may be helpful to conduct subanalysis for patients included after 2000.
- In the subanalysis regarding stone composition, what was the control for each stone type? (entire comparators or matched-pair population)
- In the individual component analysis of Mets, how were patients who initially had these components handled?
Minor points
- (line 108-109) Patients with previous components of MetS were excluded. Did it indicate the previous three or more components of MetS?
- (line 148-150) 667 stone formers had no MetS component and 71 had 1 or 2 components. what were the remaining 90 patients?
Author Response
In this study, the authors presented the risk of metabolic syndrome in kidney stone formers. Of the patients who visited their stone center, 828 were selected for the study. The control group consisted of 2484 individuals matched for age, sex, region, and follow-up period from CHIE (Care and Health Information Exchange) data. Authors reported that 43.6% of Stone formers developed metabolic syndrome, with a hazard ratio of 1.77 compared with controls. Patients with uric acid stones had a particularly high risk of developing metabolic syndrome.
The findings in this manuscript are important for the management of stone formers.
The manuscript is well written and well structured. However, I have some comments to report:
- As the authors noted, the most critical issue is that it is inaccurate to collect a component of MetS at the baseline. In their study, only 8.6% of the stone former had components of MetS (obesity, hypertension, dyslipidemia, or Impaired Glucose Tolerance). However, according to recent literature, more than 20% of patients with renal stones are reported to have hypertension (Ref 13 and 14 in their study). Obesity is common in patients with nephrolithiasis. They claimed that there was a risk of under occlusion in the control group. However, it cannot deny the possibility that many patients already had the component of MetS in Stone former at the start of the observation. If screening for MetS in GPs has been expanded after 2000, it may be helpful to conduct subanalysis for patients included after 2000.
Response: Subanalysis now included, see page 11, still remains significant
- In the subanalysis regarding stone composition, what was the control for each stone type? (entire comparators or matched-pair population)
Response: Matched pair population, clarified in text see page 10 lines 4-5
- In the individual component analysis of Mets, how were patients who initially had these components handled?
Response: Those with the specific component pre-existing were excluded, see page 10, lines 10-11
Minor points
- (line 108-109) Patients with previous components of MetS were excluded. Did it indicate the previous three or more components of MetS?
Response: Yes, granular data on the specific components was available for each patient, hence allowing for exclusion of those with pre-existing specific component in the 3rd major point.
- (line 148-150) 667 stone formers had no MetS component and 71 had 1 or 2 components. what were the remaining 90 patients?
Response: Apologies, this was a typo. Please see page 8 for amended numbers 719/828 with no components, and 111/828 with 1/2 components.
Reviewer 2 Report
Line 47-48, would make "former" plural.
Well done, good study. It is commendable to be able to get data from that long of a period and good follow up.
Author Response
Comments and Suggestions for Authors
Line 47-48, would make "former" plural.
Response: amended to 'formers'
Well done, good study. It is commendable to be able to get data from that long of a period and good follow up.
Thanks for the extremely positive and supportive comments.